# Improving Properties of Starch-Based Adhesives with Carboxylic Acids and Enzymatically Polymerized Lignosulfonates

**DOI:** 10.3390/ijms232113547

**Published:** 2022-11-04

**Authors:** Miguel Jimenez Bartolome, Sidhant Satya Prakash Padhi, Oliver Gabriel Fichtberger, Nikolaus Schwaiger, Bernhard Seidl, Martin Kozich, Gibson S. Nyanhongo, Georg M. Guebitz

**Affiliations:** 1Institute of Environmental Biotechnology, University of Natural Resources and Life Sciences, Konrad Lorenz Strasse 20, 3430 Tulln an der Donau, Austria; 2Sappi Papier Holding GmbH, Brucker Strasse 21, 8101 Gratkorn, Austria; 3Agrana Research & Innovation Center GmbH, Josef Reitherstraße 21-23, 3430 Tulln, Austria; 4Department of Biotechnology and Food Technology, Faculty of Science, University of Johannesburg, Corner Siemert and Louisa, John Orr Building, Doornfontein, Johannesburg 2028, South Africa

**Keywords:** laccase, lignosulfonates, starch, adhesive, carboxylic acid

## Abstract

A novel strategy for improving wet resistance and bonding properties of starch-based adhesives using enzymatically polymerized lignosulfonates and carboxylic acids as additives was developed. Therefore, lignosulfonates were polymerized by laccase to a molecular weight of 750 kDa. Incorporation of low concentrations (up to 1% of the starch weight) of 1,2,3,4-butanetetracarboxylic acid (BTCA) led to further improvement on the properties of the adhesives, while addition of greater amounts of BTCA led to a decrease in the properties measured due to large viscosity increases. Great improvements in wet-resistance from 22 to 60 min and bonding times (from 30 to 20 s) were observed for an adhesive containing 8% enzymatically polymerized lignin and 1% BTCA. On the other hand, the addition of citric acid (CA) deteriorated the properties of the adhesives, especially when lignosulfonate was present. In conclusion, this study shows that the addition of the appropriate amount of enzymatically polymerized lignosulfonates together with carboxylic acids (namely BTCA) to starch-based adhesives is a robust strategy for improving their wet resistance and bonding times.

## 1. Introduction

The increasing interest in substituting fossil-based products with renewable bio-based resources has made starch one of the preferred natural polysaccharides to develop various industrial applications [1,2,3]. Starch is a common granular polysaccharide of vegetable origin, composed of two different polymers of D-glucose: amylose (linear) and amylopectin (branched) [4]. Despite its advantages, as a biobased renewable resource, such as low price and wide availability, starch- properties need to be tailored to make them suitable for making specialized products. The water-sensitivity of starch [5], due to the presence of hydroxyl groups in its structure [6], make the starch-based adhesives binders less effective. This shortcoming can be solved using different chemical modifications of starch in order to incorporate hydrophobic behavior, such as acid hydrolysis, acetylation, esterification, etherification, oxidation, cross-linking and blending of polymers [7,8,9,10]. Most of said approaches use toxic compounds or do not improve the properties of the material in a sufficient way [11,12,13]. The addition of different polymers, both synthetic and biodegradable, to starch materials is a prolific topic [14], with different approaches that lead to an increase in the performance of the materials, specially the water resistance. Starch based foams have been improved using cellulose [15], fibers [16] and natural rubber latex [17], as well as poly (vinyl alcohol) [18]. Starch-based materials increased their hydrophobicity by blending them with different polymers such as poly(ε-caprolactone) [19,20], poly(L-lactide) [21] or poly(butylene succinate) [22].

Lignosulfonates would be a non-toxic and renewable alternative to increase the wet resistance of adhesives [23]. Lignin is another abundant renewable bio-based resource of vegetable origin, representing 30% of all non-fossil based organic carbon [24]. Lignosulfonates (LS), when polymerized using laccases, produces water-insoluble materials with industrial applications [25,26,27]. To avoid compromises in bonding properties upon addition of lignin, the effect of polycarboxylic acids was tested in this work.

Polycarboxylic acids are a group of safe and green chemicals whose use as additives for improving the properties of starch materials, such as retrogradation, friction coefficient, wear degree or hydrophobicity are already well known [28,29,30,31]. A wide array of polycarboxylic acids have been investigated, such as malonic acid, malic acid, tartaric acid, citric acid or succinic acid, among others [32,33]. Their effect on improving the properties of the material of the starch is expected to increase with the number of carboxyl groups in their structure [34,35]. These studies focused on starch films [30,31] and starch wood-adhesives [28,29], where the carboxylic acids citric acid (CA) and 1,2,3,4-butanetetracarboxylic acid (BTCA) were proven to be an effective strategy for improving the hydrophobicity and the retrogradation of the materials to which they were added.

Here, the combined effect of the addition of enzymatically polymerized lignosulfonates and of said carboxylic acids on starch-based paper-adhesives, in order to enhance its properties, was investigated for the first time.

## 2. Results and Discussion

### 2.1. Characterization of Laccase Polymerized Lignosulfonates

MtL8 was used to polymerize a 20% solid content LS solution, while monitoring the changes in molecular weights (MW) and viscosities. As previously reported, the viscosity of lignin increases while it is being polymerized [23,27,36]. Polymerization of LS, with a phenolic content of 247.01 mmol/mL, using the laccase led to an increase in MW from an initial average of 53.74 kDa to 750 kDa in 90 min. The increase in MW was accompanied by a steady increase in the viscosity, from an average of 5.859 mPa·s to 646.4 mPa·s at the end of the reaction (Table 1).

Laccases (benzenediol:oxygen oxidoreductases, EC.1.10.3.2) are multicopper containing enzymes that catalyze the oxidation of various aromatic compounds, reducing molecular oxygen to water. In lignin, one electron abstraction from phenolic hydroxyl groups can lead to cross-linking via formation inter-unit linkages, which include β-O-4, β-5, 5-5, β-β, 5-O-4 [37,38].

### 2.2. Viscosity

Despite the potential of enzymatically polymerized lignosulfonates to increase wet-resistance of starch based adhesive, this could be on the expense of the bonding properties [23]. On the other hand, carboxylic acids are well known and used for improving the properties of starch-based materials [29,30]. Hence, the effect of different concentrations of 1,2,3,4-butanetetracarboxylic acid (BTCA) and citric acid (CA) added to the adhesives were studied here.

When LS, polymerized until a MW of 750 KD, was present in the starch adhesive, the viscosity augmented with increasing concentration of BTCA, while the opposite occurred for the starch-only adhesives (Figure 1A). The highest BTCA concentration (80% *w*/*w*) led to a steep increase in the viscosity, changing the properties of the adhesive, turning from a viscous liquid to a paste-like adhesive, increasing more than 1150% (average of 5.14 × 10^4^ mPa·s, SD = 1.25 × 10^3^). Since the difference with the other viscosity values was very high and would make the visual comparison between the other samples difficult, it was not included in Figure 1A.

When CA was added, the viscosity of the starch-only adhesive behaved in a similar way as seen with BTCA. In this case, the addition of 10% CA decreased the viscosity of the starch-lignin adhesive, with this effect being higher with increasing amounts of this acid (Figure 1B).

The reduction on the viscosity of starch materials in the presence of CA has been previously reported in literature, due to granule surface corrosion, the hydrolysis of lamellar/nonlamellar amorphous starch as well as the degradation of starch molecules [39,40].

### 2.3. Bonding Times

In terms of the bonding time, an addition of the minimum amount of BTCA (1% *w*/*w*) led to a remarkable increase in the strength of the material, greatly reducing the time needed for a complete bonding. Interestingly, higher concentrations of BTCA had the opposite effect, being more pronounced when no lignin was present (Figure 2A).

The addition of CA led to an increase of the bonding time for both kinds of adhesives: with and without lignin. Higher concentrations of CA resulted in higher increments of the bonding times (Figure 2B).

### 2.4. Wet Resistance

A great improvement of the wet-resistance was seen for 1% BTCA added, especially when LS was present. The highest increase in wet resistance was found when 1% BTCA was added to starch based adhesives containing enzymatically polymerized lignosulfonates (MW 750 kDa). This concentration is, by far, lower than determined in other studies in which the addition of higher concentrations of BTCA (up to 80%) were reported to be optimum [29]. In this case, adding 5% BTCA does not cause any difference. Interestingly, higher concentrations of BTCA slightly deteriorated the properties of the pure starch adhesives and even had a greater negative impact when LS was present (Figure 3A).

On the other hand, the addition of CA always lowered the wet resistance of the starch only adhesive (Figure 3B), while, as seen for BTCA, the addition of 1% of the acid increased the wet resistance. The change of the properties upon addition of CA was more pronounced when lignin was present in the adhesive.

The data shows that the addition of BTCA is an attractive approach for improving the material characteristics of the adhesives, while the addition of CA had a negative effect both on bonding and only a marginal positive effect on wet-resistance. In contrast to previous reports in which high temperatures for crosslinking the acid with the starch were used [29,30,31], these beneficial effects were obtained at room temperature. Thus, most likely the improved hydrogen bonding between the acids and the starch/lignosulfonate could be responsible for the beneficial effect of BTCA [41]. The addition of LS to the starch could also increase the sites for the hydrogen bonding produced by the BTCA, which could potentially be responsible for the increase in the adhesive’s properties when both LS and BTCA are present. Hydrogen bonds are a well-known occurring phenomenon in lignin, with both the aliphatic and phenolic hydroxyl groups, with the aliphatic forming stronger bonds than the phenolic [42], as well as with the sulfonic groups of the lignin [43]. This indicates that chemical crosslinking of both materials due to the ability of the BTCA for producing highly reactive cyclic anhydrides under thermally increased conditions [44] is not necessary for improving the performance of the starch-lignin adhesives, but could be mediated by hydrogen bonding in the presence of carboxylic acids.

## 3. Materials and Methods

### 3.1. Materials and Enzymes

All used chemicals were of analytical grade. Sodium hydroxide (NaOH) was obtained from Merck (Darmstadt, Germany), 2′azino-bis (3-ethylbenzothiazoline-6-sulfonic acid) diammonium salt (ABTS) and 1,2,3,4-butanetetracarboxylic acid (BTCA) were obtained from Sigma-Aldrich (Steinheim, Germany), and citric acid was obtained from Carl Roth (Karlsruhe, Germany). Laccase from *Myceliopthera thermophila* (MtL8) was obtained from Novozymes (51003; Bagsværd, Denmark). Industrial LS from softwood (with pH 3.4; ash content of 4.4% of dry basis; solid content 17.3% *w*/*v*; reducing sugars below 100 mg/kg liquid) and the starch adhesives were provided by SAPPI Austria and AGRANA Austria, respectively.

### 3.2. Determination of Phenol Content

For determining the content of phenolic groups in the LS, the Folin–Ciocalteu (FC) method was used, as described in [45]. Briefly, the phenol groups and the FC-reagent react, forming a blue phosphotungstic–phosphomolybdenum complex and quantified by UV/VIS spectroscopy at 760 nm. A total of 20 μL LS sample and 60 μL FC-reagent and 600 μL MQ-water were incubated for 5 min at 21 °C. Subsequently, 200 μL sodium carbonate 20% (*w*/*v*) and 120 μL MQ-water were added. After 2 h at 21 °C under 800 rpm agitation, 200 μL of the reactions were transferred into a 96-well transparent plate and their absorbance measured in a plate reader (Tecan Infinite M200). A standard curve was constructed using vanillin in concentrations ranging from 0.05 to 1 mg/mL. All samples were measured in triplicate.

### 3.3. Enzymatic Polymerization of Lignosulfonates

The dried lignosulfonates (LS) were dissolved in ultra-pure water with a solid content of 20% *w*/*w*, adjusting the pH to 7. The enzymatic polymerization of 250 mL of LS solution was carried out in 500 mL transparent Durham bottles with tight-fitting tube holes on the lid to allow air supply on an IKA RCT standard magnetic stirrer equipped with an IKA ETS-D5 thermometer (IKA, Staufen, Germany). The activity of the laccase (1.761 × 10^4^ nkat/mL) was determined following the protocol described in [46]. The reaction, which was previously warmed up at 37.5 °C and continuously supplied with 10 cm^3^ min^−1^ air under constant stirring at 600 rpm, was started by introducing 233.0 nkat of MtL8 per mL of the lignosulfonate solution, corresponding to 3.300 mL of MtL8.

### 3.4. Characterization of Laccase Polymerized Lignosulfonates

In order to follow the polymerization process of the LS, two processes were carried out.

First, the viscosity increase during polymerization was monitored by rheometry. Samples of 0.8 mL were measured in an Anton-Paar (Graz, Austria) rheometer (MCR 302) with a CP50-1 cone plate at 200 1/s for 10 s with a constant temperature of 20 °C, obtaining the viscosity of the material in mPa·s.

Secondly, the molecular weight (MW) distribution was analyzed using size exclusion chromatography (SEC) following the protocol described in [13]. A total of 0.5 mL samples were withdrawn from the polymerization reaction and diluted with the mobile phase (50 mM NaNO_3_, 3 mM NaN_3_) to a concentration of 4 mg mL^−1^ before injecting 100 μL into the system. The SEC system consisted of an auto sampler 1260 series (Agilent Technologies, Palo Alto, CA, USA), a quaternary/binary pump, a Diode Array Detector and a Refractive Index detector system (Agilent Technologies 1260 Infinity), and a MALLS (Multi-Angle Light Scattering) HELEOS DAWN II detector (Wyatt Technologies, Dernbach, Germany). The system was equipped with a pre-column PL aquagel-OH MIXED Guard (PL1149-1840, 8 μm, 7.5 × 50 mm, Agilent Technologies) and a separation column PL aquagel-OH MIXED H (PL1549-5800, 4.6 × 250 mm, 8 μm, Agilent Technologies), having a mass range from 6 kDa to 10,000 kDa. The Agilent Software Openlab Chemstation CDS and the ASTRA 7 software (Wyatt technologies), were used for acquiring and processing the data, respectively. Bovine serum albumin (BSA) was used as standard for the normalization, band broadening and alignment of the MALLS detector

Samples were withdrawn at intervals and used for measurements of MW and changes in viscosity.

### 3.5. Adhesive Synthesis

Starch based adhesives were prepared following the protocol described in [11]. Briefly, 70 g of the dry starch was solubilized in 130 mL of ultra-pure water under constant stirring at 1500 rpm using an impeller (Model RZR 2020, Heidolph, Schwabach, Germany) for 45 min.

For producing the starch-lignin adhesive, laccase polymerized LS was mixed, as described above during 10 min, with the starch adhesive in a ratio of 8% *w*/*w* polymerized LS and 92% *w*/*w* starch-based adhesive.

The carboxylic acids (BTCA and citric acid) were added in order to investigate the effect on the properties of the adhesives. Different amounts of the carboxylic acid, in % solid weight of the starch, were added after the adhesive was produced. The materials were stirred as before for 10 min. The material properties of these adhesives (i.e., viscosity, wet resistance and bonding time) were measured once the acid was completely mixed and compared with samples without acid added.

### 3.6. Characterization the Adhesive

#### 3.6.1. Viscosity Measurement

The viscosity of the adhesives was measured in duplicates, with a standard deviation lower than 552 (Figure 1A) and 589 (Figure 1B), using an Anton-Paar (Graz, Austria) rheometer (MCR 302) with a CP50-1 cone plate. The sample was kept at 20 °C and stirred with an increasing shear rate from 0.1 1/s to 100 1/s.

#### 3.6.2. Bonding Measurements

For determining the bonding time, a sample of adhesive was applied in a 60 µm thick layer using a film applicator (Model 360 192495, Erichsen, Hemer, Germany) in a sheet of paper with a surface of 17 cm of length and 7 of width. Another piece of the paper, with the same surface, was put on top, pressed together by hand, and pulled apart. The time needed to produce a complete bond (i.e., the fibers of the paper tore at separation) was measured. The measurements were measured in triplicates, with a standard deviation lower than 11.3 (Figure 2A) and 17.5 (Figure 2B).

#### 3.6.3. Wet Resistance Measurement

Wet resistance was measured gluing two 20 × 3 cm stripes of paper with a 60 µm thick adhesive layer using the same film applicator previously described, overlapping an area of 6 cm of length and 3 cm of width. The samples were maintained under a light weight (660 g) overnight and kept under a controlled atmosphere in a desiccator for one week. The paper stripes were hanged with a 100 g weight in one of their ends in a container with water at 25 °C, measuring the time needed for the separation of the pieces of paper. The measurements were measured in triplicates, with a standard deviation lower than 3.66 (Figure 3A) and 1.44 (Figure 3B).

## 4. Conclusions

The addition of the carboxylic acids to a starch-lignin/lignosulfonate adhesive, was found to have a positive effect on its bonding properties and wet-resistance. However, the effects strongly depended on the acid used. The addition of BTCA improved the bonding time, reducing it by 24% when 1% of the acid was added (Figure 2A) and, at the same time, increasing by more than 175% the wet resistance of the adhesive when enzymatically polymerized lignosulfonate was present (Figure 3A). On the other hand, the addition of 1% CA, only had a marginal effect on wet resistance of 23% increment (Figure 3B) and bonding of more than 30% (Figure 2B). However, also for BTCA there is a maximum concentration where beneficial effects were seen, while at concentrations higher than 10% both bonding strength and water resistance decreased.

## Figures and Tables

**Figure 1 ijms-23-13547-f001:**
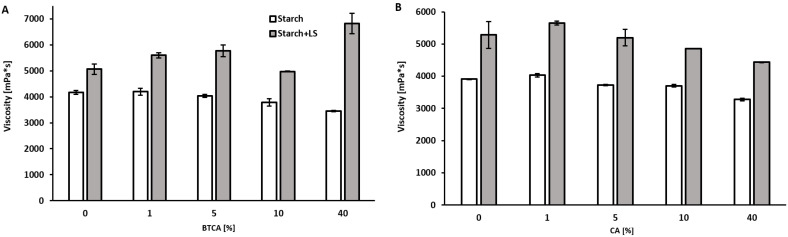
Viscosity of a starch-based adhesive with and without enzymatically polymerized LS (8%), and containing different amounts of BTCA (**A**) and CA (**B**). Error bars indicate the standard deviation (n = 2).

**Figure 2 ijms-23-13547-f002:**
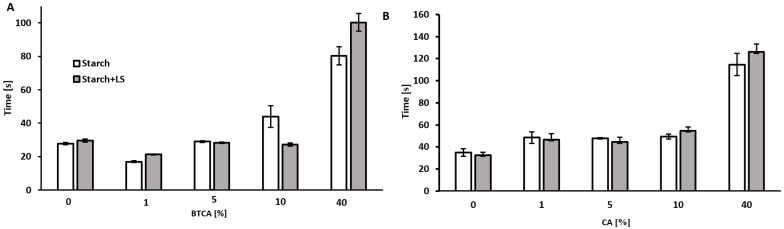
Bonding times of the starch-based adhesive, with and without enzymatically polymerized LS (8%), and containing different amounts of BTCA (**A**) and CA (**B**). Error bars indicate the standard deviation (n = 3).

**Figure 3 ijms-23-13547-f003:**
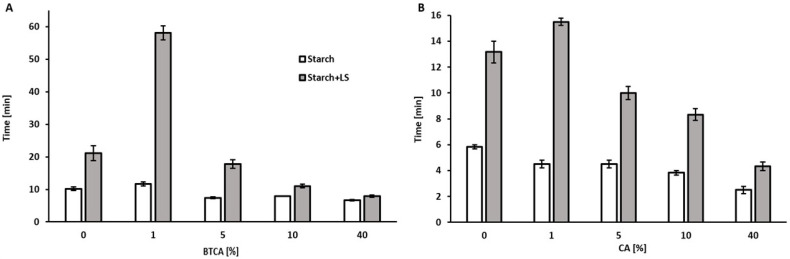
Wet resistance of the starch-based adhesive, with and without enzymatically polymerized LS, and containing different amounts of BTCA (**A**) and CA (**B**). Error bars indicate the standard deviation (n = 3).

**Table 1 ijms-23-13547-t001:** Changes in viscosity and molecular weight (MW) in different time points following the enzymatic polymerization of the lignosulfonates.

Time [min]	Viscosity [mPa·s]	Mw [kDa]
0	5.859	53.74
20	13.77	221.1
50	27.84	351.5
90	646.4	750.0

## Data Availability

The data presented in this study are available in the Appendix A.

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
