# Peer review of "Improving Properties of Starch-Based Adhesives with Carboxylic Acids and Enzymatically Polymerized Lignosulfonates"

_ijms, 2022, doi:10.3390/ijms232113547_

Round 1
Reviewer 1 Report
In the given manuscript the authors present results of Improving properties of starch-based adhesives with carboxylic acids and enzymatically polymerized lignosulfonates. Special focus is given to the influence of enzymatically polymerized lignosulfonates on the properties of starch-based adhesives.
Given data could be useful in the production of starch base adhesive with the advanced properties like bonding properties and wet-resistance improvements. In this sense some issues have to be comment in the manuscript:
- Why addition of CA has marginal effect on the wet resistance and bonding. Please discussed the differences between the effect of CA and BTCA. Please discussed synergistic effect of LS, CA and BTCA on the starch properties.
- Negative charges of the molecules should have important effect on the materials properties, according the enzymatic oxidation of lignosulfonates did you manage effect of the negative charges?
- The reaction time of the LS enzymatic polymerisation has been tested, please indicate in the Materials and methods section how reaction was stopped.
- Please discussed and propose the mechanism of the LS enzyme polymerisation
- Please provide data of Laccase activity measurements results. Did you observe reduction in activity according the changes of the viscosity?
- Please provide more details about the enzymatic polymerisation experiments e.g. mixing, temperature regulation, pO2? These parameters are important for the laccase activity.
- Enzyme amount used in this study is not clear.
- In the section Materials and Methods/Enzymatic polymerisation of lignosulfonates: durum bottle? Durham bottles?
- In the section Materials and Methods/Adhesive synthesis: parentheses have to be given for the reference 11.
- Figure 3 x-axis change BCA to BTCA.
- In the section Results and discussion/Wet resistance: “In contrast to previous reports, in which high temperatures for crosslinking the acid with the starch were used these beneficial effects were obtained at RT.” Please indicate meaning of the abbreviation RT.
Reviewer 2 Report
Improving properties of starch-based adhesives with carboxylic acids and enzymatically polymerized lignosulfonates.
ijms-1918703-peer-review-v1
Summary
The study on improving properties of starch-based adhesives with carboxylic acids and enzymatically polymerized lignosulfonates has been done thoroughly. The work is novel specifically in terms of combined effect of the addition of enzymatically polymerized lignosulfonates and of said carboxylic acids on starch-based paper-adhesives. Results has been discussed with supporting data. A few modifications are suggested before publishing the article.
Comments:
· Lines 36- 38: The different chemical modifications can be used to reduce the water absorption of starch. Refer them and cite appropriately. (References: https://doi.org/10.1007/s10570-021-04199-6 and https://doi.org/10.3390/molecules26226880)
· Other than pure starch-based adhesives, starch-PVA adhesives and starch compounded with other biodegradable polymers can also be produced. The strategies that can be used to increase the water-resistant properties may be included in the following article and cite where necessary.
https://doi.org/10.3390/jcs5110300
https://doi.org/10.1021/acsomega.2c01292
Introduction should be bit improved. Author should discuss what people have done previously, bit discuss their finding, what missing by identifying research gap, and why your proposed material(s) are important and novelty.
Keep the space for number and unit, for instance 660g 20oC etc, you should check entire manuscript
Section 2 and 3 should be changed, Materials and method should appear first before results and discussion

Reviewer 3 Report
In summary, this is a weak manuscript, with no statistical testing and the underpinning chemistry is not at all described. There are too few figures and I recommend it not to be published.
